# Effect of Trace Rare-Earth Element Ce on the Microstructure and Properties of Cold-Rolled Medium Manganese Steel

Qingbo Zhao [1,2], Ruifeng Dong [1,3,*], Yongfa Lu [1], Yang Yang [1], Yanru Wang [1] and Xiong Yang [4]

1   School of Materials Science and Engineering, Inner Mongolia University of Technology, Hohhot 010051, China
2   Beijing BOE Display Technology Co., Ltd., Beijing 102600, China
3   Engineering Research Center of Rare Earth Metals, Inner Mongolia University of Technology, Hohhot 010051, China
4   Technical Center of Inner Mongolia Baotou Steel Union Co., Ltd., Baotou 014010, China
*   Correspondence: drfcsp@163.com

**Abstract:** Rare-earth elements have been widely used in the field of functional materials, but their effects on the cold-stamping formability of high-strength automotive steels have rarely been studied. In this paper, the effect of the trace rare-earth element Ce on the microstructure and properties of cold-rolled medium manganese steel after ART (austenite-reverted transformation, ART) annealing was studied. The microstructure of the experimental steel was observed using SEM, and the mechanical properties were tested using a universal tensile testing machine. The volume fraction of the retained austenite and the texture of the steel were measured using XRD. The results showed that the original austenite grain size of the experimental steel was smaller after adding the trace rare-earth element Ce. After ART annealing, the grain size distribution of the experimental steel with rare-earth Ce was more uniform, and the comprehensive mechanical properties were better. Under the conditions of quenching at 800 °C for 5 min and annealing at 645 °C for 15 min, the maximum product of tensile strength and elongation was 28.47 GPa·%.

**Keywords:** trace rare-earth element; cold-rolled medium manganese steel; original austenite; microstructure; properties





## 1. Introduction

In recent years, the third generation of automobile steel, represented by medium manganese steel (Mn content of 3–10%), has gradually become the main research hotspot at home and abroad due to its excellent high-strength and high-plasticity mechanical properties [1]. The design of alloy composition is an important factor affecting the final properties of materials and has been extensively studied [2] As a strategic resource in China, rare earth (RE) is widely used in the field of functional materials, such as high-performance permanent magnet materials and luminescent materials, but is used less in metal structural materials [3].

Some scholars have studied the effects of rare-earth elements on steel, such as Guo Feng et al. [4], who found that rare earth can improve the morphology of inclusions, purify grain boundaries, improve the strength of grain boundaries, reduce the possibility of crack propagation through defects, and improve impact toughness. Zhao et al. [5] found that the addition of rare earth increased the AC1 and AC3 phase transition temperatures of low-carbon microalloyed high-strength steel, reduced the critical cooling rate of the pearlite transformation, increased the incubation period, and made it easier to obtain a bainite structure. Liu Chengjun et al. [6] pointed out that rare earth can refine the austenite grain boundary to improve impact toughness. QU et al. [7] studied the second phase in rare-earth HSLA steel and found that the average size of precipitates was refined by about 15 nm after adding rare earth, indicating that rare earth can promote the precipitation of fine carbon and nitride, which is beneficial for improving the strength and toughness of materials.

At present, the effect of rare-earth elements on medium manganese steel is not clear, and the role of rare earth in steel is not clear [8–10]. How to reasonably and effectively use rare-earth elements in steel remains to be further studied.

The selection of rare-earth content is also critical to the study of the microstructure and properties of steel. Considering the internal quality of the material, adding a large amount of rare-earth elements produces large inclusions in the structure, which affects the plasticity and performance of the material and deteriorates the forming performance of automobile steel. From the perspective of production practice, automobile steel is mostly plate, and a large number of rare-earth elements are added. During the casting process, large particle inclusions block the casting nozzle and cause casting accidents. This study selected trace rare-earth elements for testing and research.

In this paper, cold-rolled medium manganese experimental steels with 9 ppm rare-earth Ce and without rare earth were selected for comparative study. The two groups of experimental steel were subjected to the ART annealing treatment, and their microstructures, properties, and textures were analyzed to explore the effect of the trace rare-earth element Ce on the microstructure and properties of cold-rolled medium manganese steel, which provided a theoretical basis for the research and development of new third-generation rare-earth automotive steel, which is very economically significant for the development of the rare-earth steel and automotive fields.

## 2. Experimental Materials and Methods

The experimental steel was melted and cast by a laboratory vacuum induction furnace, and the ingot size was 100 mm × 100 mm × 300 mm. Then, a 1 mm thick experimental steel plate was obtained by rolling. The chemical composition of the test steel is shown in Table 1. The phase transformation temperature of the experimental steel was measured by a NETZSCH STA449 F3 comprehensive thermal analyzer at a rate of 20 °C/min to 950 °C under argon protection. The $A_{C1}$ and $A_{C3}$ values of the experimental steel without rare earth were 596 °C and 734 °C, respectively. The $A_{C1}$ and $A_{C3}$ values of the rare-earth experimental steel were 598 °C and 751 °C, respectively. The heat treatment process was conducted as shown in Figure 1. The experimental steel plate was quenched at 800 °C for 5 min and then annealed at 625 °C, 645 °C, or 665 °C for 15 min.

**Table 1.** Chemical composition of test steel (%).

| Steel | C | Mn | Si | Al | Cu | Ni | Nb | Ti | P | S | Ce |
|---|---|---|---|---|---|---|---|---|---|---|---|
| 0 RE | 0.089 | 4.873 | 0.113 | 0.047 | 0.28 | 0.263 | 0.030 | 0.028 | 0.011 | 0.006 | - |
| 9 ppm RE | 0.093 | 4.899 | 0.128 | 0.071 | 0.26 | 0.246 | 0.027 | 0.029 | 0.011 | 0.006 | 0.0009 |

The original austenite corrosion was carried out on the quenched experimental steel, and the supersaturated picric acid solution was used as the corrosive agent. The metallographic structure after corrosion was observed, and the original austenite grain size distribution was determined using Image Pro Plus software [11]. The small $10 \times 10 \times 1$ mm$^3$ pieces were cut from the annealed steel plate as a microstructure observation sample. The samples were polished and corroded with 4% nitric acid alcohol. The microstructures were observed using a field emission scanning electron microscope. The tensile specimens were prepared according to the GB/T 228.1-2010 standard. The length direction was parallel to the rolling direction. The original gauge was calculated by the proportional gauge ($L_0 = 5.65\sqrt{S_0}$), and the tensile test was carried out on a universal tensile testing machine. Two groups of experimental steels were tested using an X-ray diffractometer (XRD, X Pert PRO MPD). The sample size was 10 mm × 15 mm. The (200), (220), (311) crystal plane diffraction peaks of the FCC phase and the (200) and (211) crystal plane diffraction peaks

of the BCC phase were measured, and the reverse-transformation austenite content was calculated using the following formula:

$$Vi = \frac{1}{1 + G(I\alpha / I\gamma)} \tag{1}$$

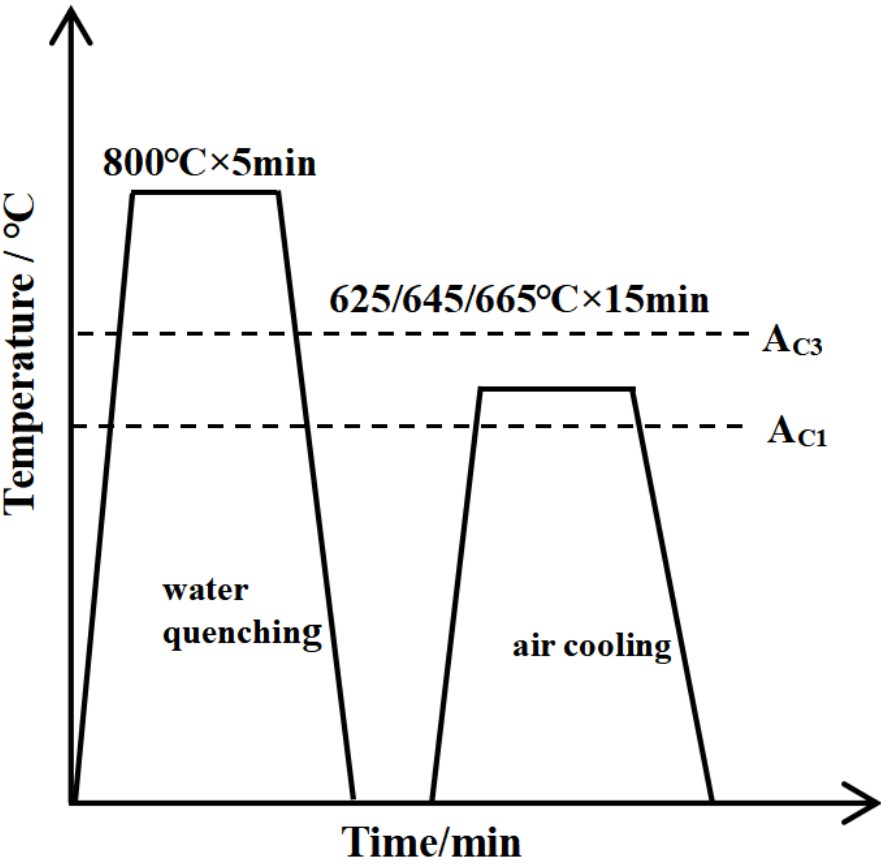

**Figure 1.** Heat treatment process of test steel.

In the formula, *G* represents the lattice parameter correlation value of the FCC phase and BCC phase [12], and the detailed correspondence is shown in Table 2. *Vi* is the volume fraction of reversed austenite corresponding to different *G* values, *Iγ* is the integral intensity of the (200), (220), and (311) crystal plane diffraction peaks of the FCC phase, and *Iα* is the integral intensity of the (200) and (211) crystal plane diffraction peaks of the BCC phase. The texture of the samples was also measured using an X-ray diffractometer. The ODF cross section was characterized using the Roe method, and the content of each texture component was calculated using the texture analysis software ResMat-TexTools.

**Table 2.** G values corresponding to different crystal face indices.

| G-Value | (200)γ | (220)γ | (311)γ |
|---|---|---|---|
| (200)α | 2.46 | 1.32 | 1.78 |
| (211)α | 1.21 | 0.65 | 0.87 |

## 3. Results and Analysis

### 3.1. Original Austenite Grains

According to the measured phase transition temperature, it was found that the $A_{C3}$ temperature of the experimental steel increased after adding rare-earth elements. To further verify this point, the two groups of experimental steels were subjected to original austenite corrosion to observe the changes in the original austenite grains. Figure 2 shows the original austenite metallographic structure and grain size distribution. It can be seen in panels (a) and (b) that the austenite grains of the experimental steel with rare-earth Ce were smaller. The increase in $A_{C3}$ temperature by 17 °C indicates that the austenite grains nucleated first in the austenitizing process of the experimental steel without RE elements, while the austenite grains nucleated later in the experimental steel with RE elements. Therefore, after the quenching treatment, the original austenite grains in the experimental steel containing rare earth were smaller [11]. The average grain sizes of the original austenite of the two groups of experimental steels were measured using Image-Pro Plus software to be 3.3 μm and 2.5 μm, respectively. The grain size distribution is shown in panels (c) and (d). The grain size of rare-earth experimental steel was small, where the grains between 2 and 4 μm accounted for about 43.91% and the grains less than 2 μm accounted for about 40.92%. The grain size of experimental steel without rare earth was mainly concentrated in 2~4 μm (about 56.23%). It can be seen that with the addition of trace rare-earth elements the austenite grains in the experimental steel were refined and the microstructure became more uniform.

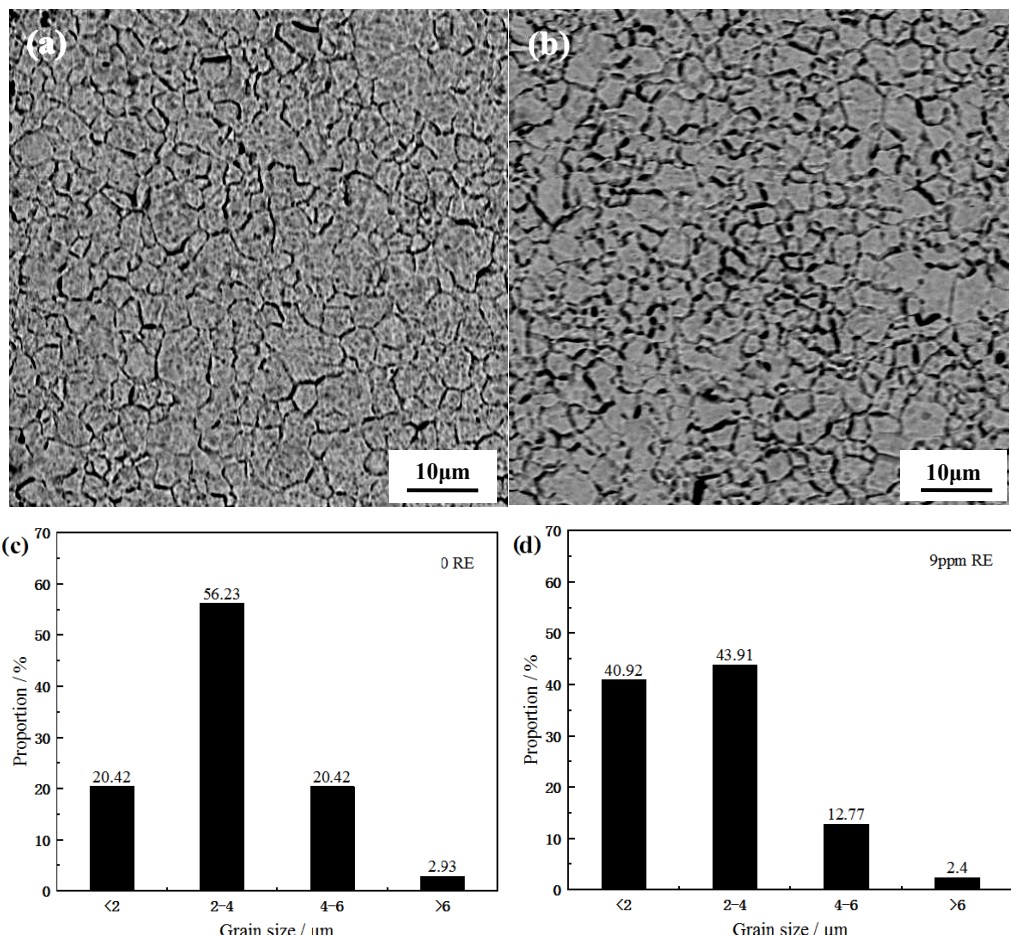

**Figure 2.** Original austenite microstructure and grain size distribution of two groups of experimental steels: (**a**,**c**) 0 RE and (**b**,**d**) 9 ppm RE.

*3.2. Microstructural Evolution*

Two groups of experimental steels were annealed by ART, and the microstructure after annealing was observed. The results are shown in Figure 3. It can be seen from the diagram that the microstructure after annealing was mainly composed of ferrite and retained austenite (or new martensite), in which the protruding structure was austenite or new martensite and the concave structure was ferrite [13]. Panels (a) and (d) show the microstructure at 625 °C. A large number of white granular carbides were dispersed in the microstructure. Panels (b) and (e) show the microstructure at 645 °C. It can be seen that there were still many white granular carbides on the ferrite [14], while there were almost no white granular carbides on the protruding austenite structure. This was because the annealing temperature was low at this time, the carbides had not been completely dissolved during the growth of austenite, and these carbides were attached to the ferrite. Panels (c) and (f) show the microstructure of the experimental steel at 665 °C. At this time, the carbides had completely disappeared, and the microstructure was composed of ferrite, austenite, and a small amount of new martensite. The austenite grains in the structure grew, and there were two forms, which were the lath and block distributions in the structure. This was also because the austenite grains grew with the increase in temperature, and some larger austenite was transformed into martensite during cooling due to the lower stability of carbon and manganese per unit volume.

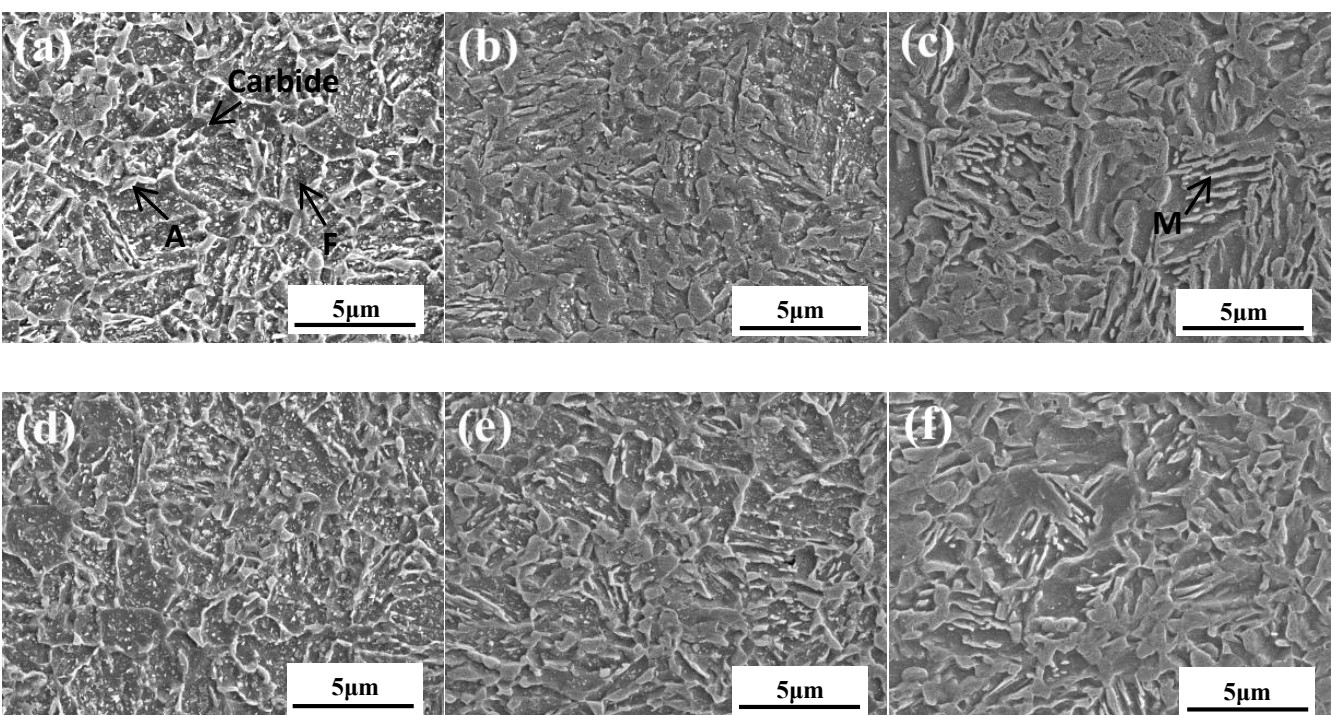

**Figure 3.** Microstructure of two groups of experimental steels at different annealing temperatures: (**a**,**d**) 625 °C, (**b**,**e**) 645 °C, (**c**,**f**) 665 °C; (**a–c**) 0 RE, (**d–f**) 9 ppm RE.

Through the comparative analysis of the microstructure of the two groups of experimental steels under different annealing processes, it was found that the grain size of the experimental steel with trace rare-earth elements was smaller. There were two reasons for this. On one hand, rare-earth elements themselves have the effect of grain refinement. The radius of rare-earth atoms is about 1.5 times that of Fe atoms, which can only dissolve into defects in the crystal but cannot dissolve into austenite to form a solid solution. Because there are many defects at the grain boundary, it is easy to accumulate rare-earth atoms near the grain boundary, which hinders the diffusion of atoms, thereby inhibiting the growth of austenite grains and achieving the effect of grain refinement [15]. On the other hand,

rare-earth elements have the effect of deoxidation and desulfurization, which can produce fine and stable rare-earth oxides, rare-earth sulfides, or oxygen sulfides. These rare-earth compounds can be used as heterogeneous nuclei to provide excellent conditions for the refinement of crystallization.

### 3.3. Residual Austenite Content

The experimental steel was tested using XRD after different heat treatment processes, and the results are shown in Figure 4. In the figure, γ represents the FCC phase, α represents the BCC phase, and the volume fraction of austenite was calculated according to the integral strength of the austenite and ferrite diffraction peaks. As can be seen in the figure, with the increase in the annealing temperature, the austenite diffraction peak was enhanced. When the annealing temperature was 645 °C, the austenite content was the highest. When the temperature continued to increase, the diffraction peak of the FCC phase obviously decreased, while that of the BCC phase increased, indicating that the austenite content decreased with the increase in the annealing temperature, and new martensite was formed in the matrix structure. The FCC phase diffraction peak of the experimental steel containing rare earth was significantly higher than that of the experimental steel without rare earth. It can be seen that the addition of trace rare-earth elements increased the volume fraction of the retained austenite in the steel.

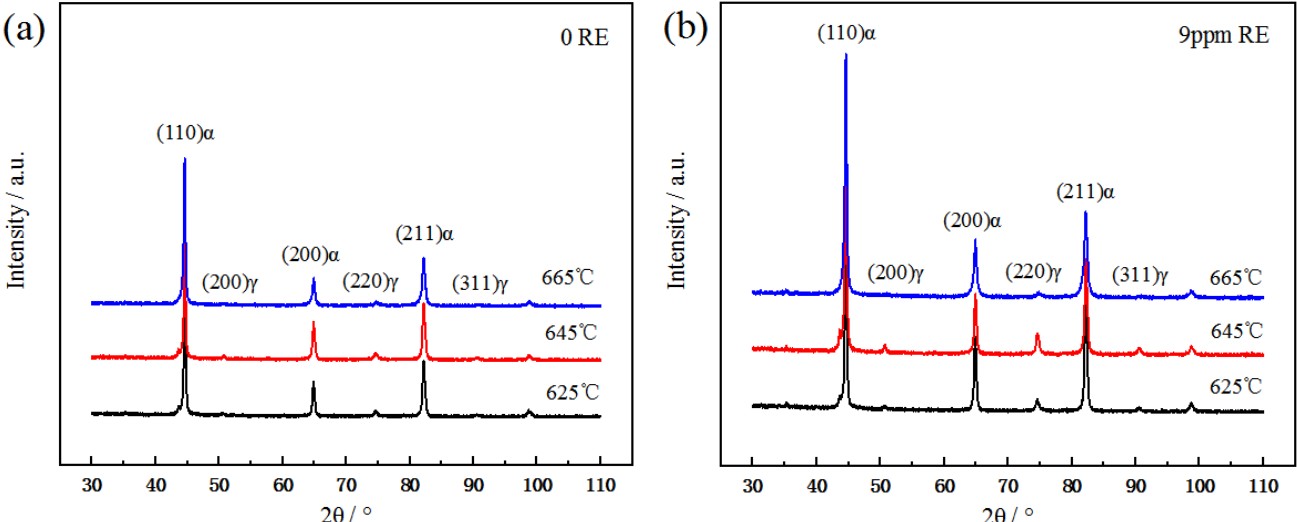

**Figure 4.** XRD patterns of the experimental steel at different annealing temperatures: (**a**) 0 RE, (**b**) 9 ppm RE.

Figure 5 shows the calculated volume fractions of the retained austenite of the two groups of experimental steels after different annealing processes. It can be seen in the figure that with the increase in temperature, the content of retained austenite increased first and then decreased. This was due to the unstable growth of austenite grains in the microstructure at higher annealing temperatures; the insufficient distribution of C and Mn elements, leading to the low contents of C and Mn elements per unit volume of austenite; and the transformation tendency of martensite to increase during high-temperature annealing, so some austenite transforms into martensite during the cooling process [16]. When the experimental steel was annealed at 645 °C, the austenite content was the largest, and the residual austenite contents of the two experimental steels were 21.1% and 22.8%, respectively. In comparison, it is found that adding trace rare-earth elements to steel was beneficial for retaining more austenite at room temperature.

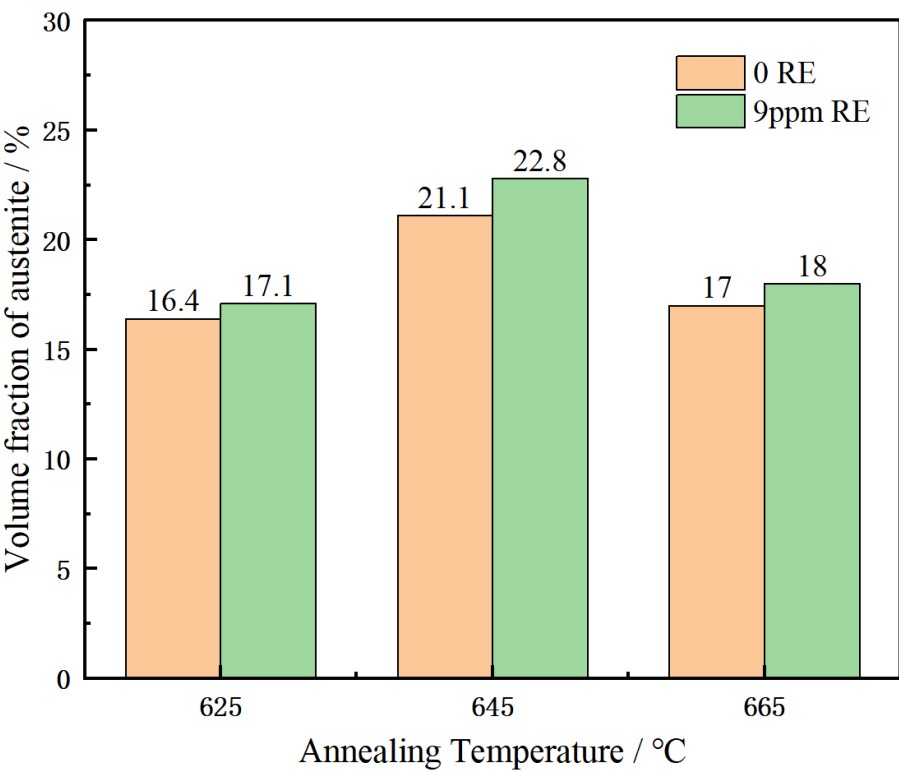

**Figure 5.** Residual austenite contents of the experimental steels at different annealing temperatures.

*3.4. Mechanical Properties*

Figure 6 shows a comparison of the mechanical properties of the two groups of experimental steels after the ART annealing treatment. It can be seen in the figure that the yield strength and tensile strength of the two groups of experimental steels after annealing were very similar, but the mechanical properties of the experimental steel with the rare-earth element Ce were improved. The yield strength decreased with the increase in the annealing temperature, and the maximum value was 850 MPa at 625 °C. The tensile strength increased with the increase in the annealing temperature, and the maximum value was 1100 MPa at 665 °C. The elongation increased first and then decreased with the increase in the annealing temperature and reached a maximum at 645 °C. The elongation of the experimental steel without rare earth was 27.82%, while the elongation of the experimental steel with rare earth was 33.89%. The change rule of the product of strength and elongation was consistent with that of elongation, which increased first and then decreased. The maximum value is obtained when the annealing temperature reached 645 °C. At this time, the products of strength and elongation of the experimental steels without rare earth and with rare earth were 24.3 GPa·% and 28.47 GPa·%, respectively. It can be seen that the rare-earth element was beneficial for the improvement of the mechanical properties of the experimental steel, and 800 °C for 5 min and 645 °C for 15 min was the best heat treatment condition for the experimental steel.

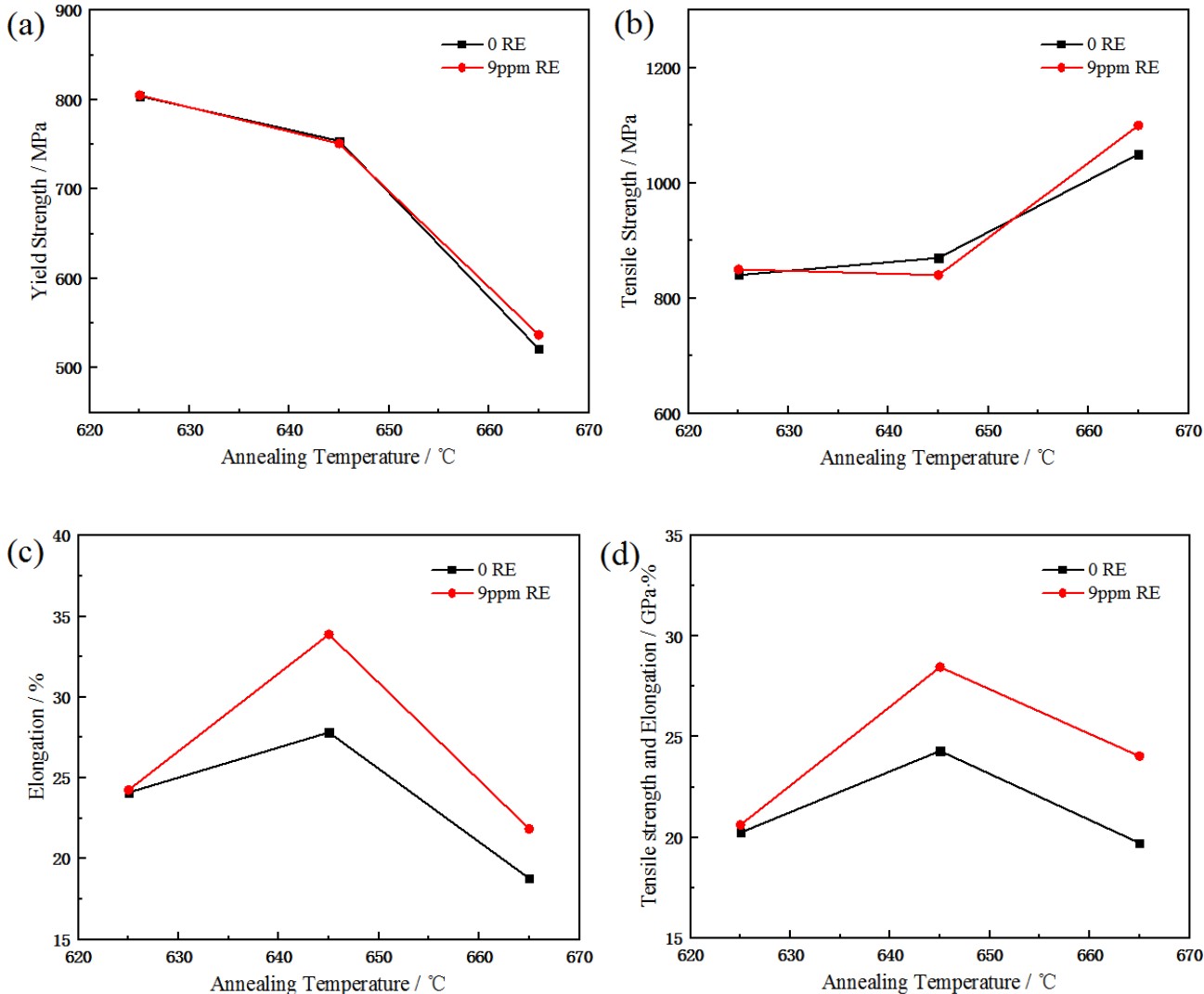

**Figure 6.** Mechanical properties of two groups of experimental steels at different annealing temperatures: (**a**) yield strength, (**b**) tensile strength, (**c**) elongation, and (**d**) the product of strength and elongation.

The change in the product of strength and elongation was the same as that of the austenite content after annealing at different temperatures, which indicates that the austenite content is an important factor affecting the comprehensive mechanical properties of experimental steel. Combined with the analysis of the XRD results, the reason for this change in the experimental steel may be that at a lower annealing temperature there are carbides in the microstructure and the C element in the matrix does not diffuse on a large scale, that is, it is enriched in the reverse-phase-transformation austenite. At the same time, the volume fraction of austenite is low and the stability is strong at this temperature. The comprehensive effect of the second-phase strengthening of carbides and the solid-solution strengthening of the C element makes the tensile strength and yield strength of the experimental steel remain within a certain range. With the increase in annealing temperature, the carbides in the microstructure continue to dissolve until they disappear, and the second-phase strengthening effect in the alloy also decreases and disappears. When the temperature is high, the microstructure of the experimental steel is transformed into ferrite, austenite, and martensite multiphase structures. The volume fraction of reversed austenite decreases and becomes unstable, and the TRIP (transformation-induced plasticity, TRIP) effect is weakened accordingly. During the cooling process, the hard-phase martensite increases. The higher the temperature, the more the martensite is transformed by cooling,

so the tensile strength increases and the elongation decreases. Some scholars have also put forward the same conclusion that increasing the temperature increases the austenite content, but the austenite stability is poor, which is due to the tendency for martensite transformation in high-temperature annealing and the coarsening of austenite, so it is partially transformed into martensite during cooling and the tensile strength increases [17].

### 3.5. Texture Analysis

Good stamping properties are also an important goal in the field of automobile manufacturing. Deep drawability is related to the composition of a favorable texture in the steel. The larger the {111}/{100} coefficient of the favorable texture, the larger the r-value, indicating that the deep drawability of the steel plate is better [18]. Therefore, the essence of the study of the stamping properties of steel plates is to obtain more favorable textures in the microstructure. The type and density distribution of related textures can be determined with the orientation distribution function (ODF). The denser the orientation lines in the ODF cross section, the greater the orientation density. However, in general, only the main orientation distribution changes in the ODF diagram need to be analyzed. During the rolling process of steel plate, the grain orientation gradually converges to the $\alpha$ orientation line and the $\gamma$ orientation line. Therefore, we mainly observed the texture type and density ($\varphi_2 = 45°$) on these two orientation lines.

Figure 7 shows the textured ODF cross sections of two groups of experimental steels at the optimal annealing temperature of $\varphi_2 = 45°$. It can be seen that there were {001}<110>, {011}<110>, {111}<110>, and {111}<112> textures in the two experimental steels after holding at 645 °C for 15min, and the texture density levels of the two experimental steels were very close. The strong peak position of the two experimental steels was near the {111}<110> texture on the $\gamma$ orientation line, but the texture distribution of the experimental steel with the rare-earth element Ce was more uniform, and the texture density was up to 2.6. The contents of each orientation texture of the two groups of experimental steels were compared, as shown in Table 3. It can be seen that the contents of each orientation texture of the experimental steels increased slightly after adding rare-earth elements. Among them, the content of the {111} texture increased more. Combined with the analysis of Figure 7, the trace rare-earth elements may affect the texture of the experimental steel, which is beneficial for improving the forming properties, but the effect is not obvious compared with the effect on the mechanical properties. This requires further analysis of the changes in the favorable texture {111} and unfavorable texture {100} components.

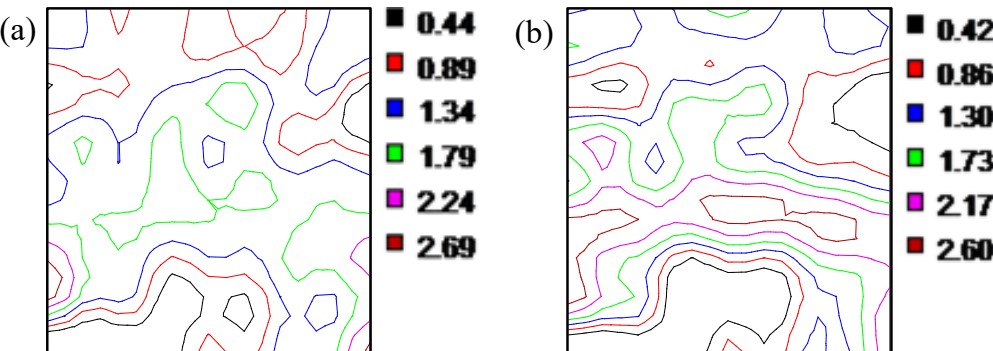

**Figure 7.** ODF cross sections of $\varphi_2 = 45°$ of two groups of experimental steels under different heat treatment processes: (**a**) 0 RE, (**b**) 9 ppm RE.

**Table 3.** Texture contents of each orientation after the ART annealing process.

| ART Annealing | {001}<110> | {112}<110> | {223}<110> | {111}<110> | {111}<112> |
|---|---|---|---|---|---|
| 0 RE | 2.5 | 4.8 | 5.3 | 4.6 | 3.2 |
| 9 ppm RE | 2.9 | 5.0 | 5.4 | 4.9 | 4.4 |

The {111} texture of the steel sheet is beneficial for the improvement of stamping performance, the {100} texture is not conducive to the improvement of stamping performance, and the {110} texture is located between the two textures. The texture contents of the {111}, {110}, and {100} planes of the two groups of experimental steels under two heat treatment processes were analyzed and calculated using ResMat-TexTools software, as shown in Figure 8. In the figure, it can be seen that the contents of the favorable textures {111} and {110} in the experimental steel with rare earth were higher than those in the experimental steel without rare earth. The content of the {111} texture was 13.4%, the content of the {110} texture was 19.6%, and the content of the unfavorable texture {100} was only 11.1%. Thus, the addition of trace rare-earth elements is beneficial for improving the formability of steel.

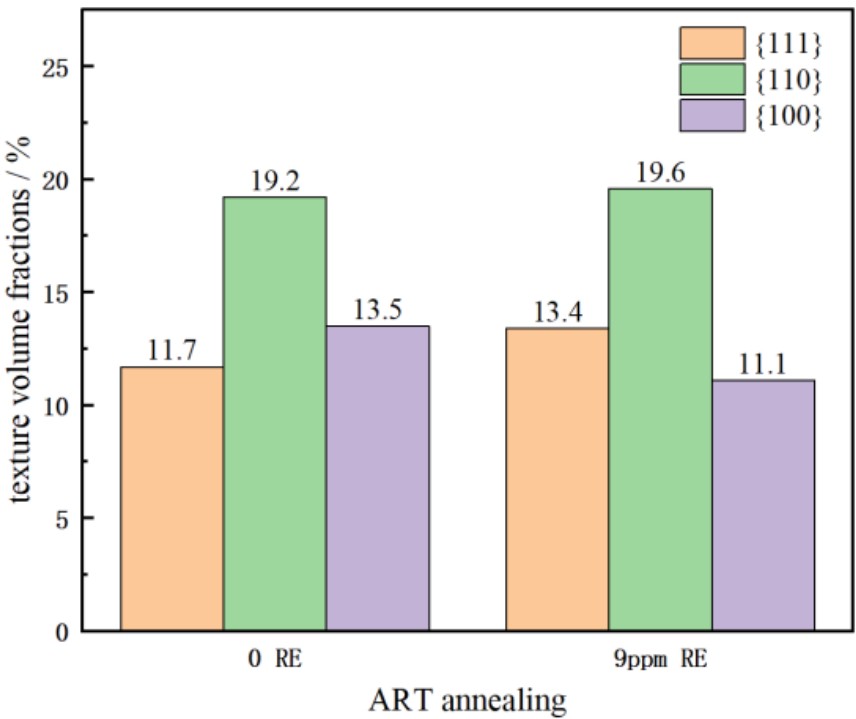

**Figure 8.** Texture contents of two groups of experimental steels after different annealing processes.

## 4. Conclusions

(1) After adding trace rare-earth Ce, the $A_{C3}$ temperature of the experimental steel increased, which delayed the nucleation of austenite in the microstructure of the rare-earth experimental steel during quenching, resulting in smaller original austenite grain sizes.

(2) The grain size distribution of the experimental steel with 9 ppm RE was more uniform, the residual austenite content was higher, and the mechanical properties were better than those of RE-free steel. The maximum product of the strength and elongation of the experimental steel with 9 ppm RE was obtained when quenching at 800 °C for 5 min and annealing at 645 °C for 15 min. At this time, the residual austenite content of the experimental steel was 22.8%, the tensile strength was 840 MPa, the elongation was 33.89%, and the product of strength and plasticity was 28.47 GPa·%.

(3) The texture distributions and density levels of the two groups of experimental steel were similar after adding trace rare earth, but the volume fraction of the favorable texture {111} increased and the volume fraction of the unfavorable texture {100} decreased. It can be seen that the addition of trace rare-earth elements can improve the microstructure of steel and improve the comprehensive mechanical properties.

**Author Contributions:** Methodology, R.D. and X.Y.; validation, Q.Z., Y.L., Y.Y. and Y.W.; writing—original draft preparation, Q.Z.; writing—review and editing, Q.Z. and R.D.; funding acquisition, R.D. All authors have read and agreed to the published version of the manuscript.

**Funding:** This research was funded by [Advanced Education Research Project in Inner Mongolia Autonomous Region] grant number [NJZZ18076], [Inner Mongolia Autonomous Region Science and Technology Plan Project] grant number [2021GG0238], [2021 undergraduate college-level 'College Students' Innovation and Entrepreneurship Training Program] grant number [2022044059] and [Inner Mongolia University of Technology 2022 Undergraduate Science and Technology Innovation Fund Project] grant number [27].

**Data Availability Statement:** Data is contained within the article.

**Conflicts of Interest:** The authors declare no conflict of interest.

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
