# Peer review of "Effect of Trace Rare-Earth Element Ce on the Microstructure and Properties of Cold-Rolled Medium Manganese Steel"

_metals, doi:10.3390/met13010116_

Round 1
Reviewer 1 Report
The paper deals with the characterisation of the cold rolled manganese steel with only 0.0009% cerium as rare earth element.
It had been reported that a suitable content of rare earth elements is 0.2-1%wt (see ref [9] provided in the article).
The authors need to provide further proof (either experimental or from other works) that the trace amount of cerium is responsible for the significant processes described in the paper.
Please see attached.

Author Response
In this study, 9ppm rare earth elements were added, mainly from the following two aspects : On the one hand, considering the internal quality of the material, Adding a large number of rare earth elements in the structure will produce large inclusions, which will affect the plasticity and toughness of the material, thereby deteriorating the formability of automotive steel; On the other hand, considering from the production practice, automobile steel multi-purpose plate, adding excessive rare earth elements, in the casting process, large particle inclusions will block the casting nozzle, resulting in broken casting accidents, so this experiment selected trace rare earth elements for testing and research. Now will reply to the views one by one :
C1: The references in the article do have improper references, which have been modified.
C2: A test method for the Ac3 transition temperature has been written.
C3: ART initials have been explained in the text.
C4: The steel will produce second phase precipitates during heat treatment. This article confirms that the size of the precipitates decreases after the addition of rare earth elements in the steel, thereby improving the toughness of the steel.
C5: The determination of the organization is indeed unconvincing. In this paper, the determination of the phase in the microstructure is based on previous research and other people 's research results, which has been cited in the text.
C6: The determination of ferrite, martensite, etc. in the structure is based on previous studies and is cited in the text.
C7: It has been cited in the text.
C8: Due to the difference of trace elements, the tensile strength does not differ much, so we take the comprehensive mechanical properties as the basis for judgment, that is, strong plastic product.
C9: It has been cited in the text.
C10: It has been modified in the text.
C11: It has been cited in the text.
C12: After adding rare earth elements, the tensile yield strength of steel has little improvement, but its comprehensive mechanical properties have improved.
C13: The steel obtained the best mechanical properties under the heat treatment conditions of 800℃-5min/645℃-15min.

Reviewer 2 Report
The effect of rare earth element Ce on microstructures and properties of a cold rolled medium manganese steel was investigated in this study. The addition of Ce produced a grain refinement effect, which hence resulted in better mechanical properties. However, there are several issues that need to be addressed in this paper.1. The introduction can be separated into several paragraphs to better present research background, gaps and strategies.
2. There are typos in some sentences, such as Line 74 on Page 2. Please go through the manuscript to emend those typos.
3. The black fonts in Figure 3 are defocusing and invisible. Please improve it.
4. The error bars are missing in Figure 6.
5. It is suggested to add a section where the mechanism involved in this process could be discussed.
Author Response
Inappropriate places such as the introduction and the typos in the text have been checked and modified in the text.
Reviewer 3 Report
The article presents the results of research on the effect of rare earth element additions Ce on the microstructure and strength properties of manganese steel. The subject of the article is current and raises an interesting issue related to the possibility of influencing the microstructure and properties of steel widely used in the automotive industry.
Recently, articles on rare-earth steels have appeared, so it would be reasonable to explain in detail in the introduction what is new in the presented article.
Other notes:
1. On what basis was the content of rare earth elements Ce in the tested steel assumed? Were studies conducted on the effect of Ce content in steel on its properties and microstructure?
2. Has the effect of rare earth additions on unalloyed steels been investigated as well? Will the observed phenomenon also occur for other types of steel?
3. What can explain such a large change in the elongation of steel with the addition of rare earth elements?
Author Response
- Considering the selection of rare earth content from two aspects. On the one hand, considering the internal quality of the material, adding a large number of rare earth elements in the structure will produce large inclusions, thereby affecting the plasticity and toughness of the material, thereby worsening the formability of automotive steel ; on the other hand, from the production practice, automotive steel multi-purpose plate, adding excess rare earth elements, in the casting process of large particle inclusions will block the casting nozzle, resulting in broken casting accident, so this experiment selected trace amounts of rare earth elements for testing. The final actual rare earth content is obtained by ICP test. The effect of trace rare earth elements on the microstructure and properties of steel was analyzed by comparing with the experimental steel without rare earth.
- The effect of rare earth elements on non-alloyed steel remains to be further studied, and related experiments are planned.
- After adding rare earth elements, the content of retained austenite in the microstructure of steel increases after heat treatment, so that the elongation increases.

Round 2
Reviewer 1 Report
The authors had addressed many of the aspects presented in the previous review. I have the following comments on the current manuscript:
C1: Although the authors presented the reasoning for choosing the low content of the cerium rare earth, it does not transpire to the reader of the manuscript these reasons. I recommend conveying the justification set forth by the authors by appending a paragraph in the introduction section with the reason to choose trace amount of cerium, as they had previously replied to the issue. Any reference of previous works would add a positive precedent for the approach proposed herein.
C2: In Figure 4, the authors use 2θ (“theta”) symbol for XRD scattering angles. In Figure 7, and in text, the authors should use the same (theta) symbol for 2φ=45° (not the „phi” symbol).
C3: This review was made with the reserve that I could not find references [3], [5], [6], [9],[10], [11], [13], [14], [15], [16], [17]. However, the authors should make their due diligence in providing adequate references. Issues that affect the quality of the references relate, but may not be limited to:
· the nature of publishing media, as it is only “Inner Mongolia University of Technology” for references [3], [10], [15], [16]. It is essential for the authors to include the journal or book title and pagination in the bibliography for references [3], [10], [15], [16].
· repeating of references, such as [11] and [13].
· wrong journals and/or issues, such as for references [11] and [13], where references seem to be cited from “Journals of Rare Earths” (not Rare Earths, as stated), but with the wrong volume. The latest volume (i.e. 2023) is number 41, while the authors cite volume 42.
After the above issues are thoroughly addressed, I recommend the article for publishing.
Author Response
Thank you to the reviewer for your valuable comments, we have revised the article according to your comments:
C1 In the introduction part, the selection of trace rare earth is described.
C2 The representation of the diffraction angle has been modified in this paper. Due to the differences in analysis methods and angles, the representation of the diffraction angle is also different.
C3 The journals that can 't be found may be because they are Chinese journals, which we have revised. In addition, references that are repeated or inappropriate are also modified.